# Migration and its impact on universal HIV testing and treatment in the HPTN 071 (PopART) study communities

David Macleod[1]*, Sian Floyd[1], Kwame Shanaube[2], William Probert[3], Justin Bwalya[2], Ab Schaap[2], Timothy Skalland[4], Ayana Moore[5], Estelle Piwowar-Manning[6], Graeme Hoddinott[7,8], Virginia Bond[2,9], Musonda Simwinga[2], Nomtha Mandla[7], Deborah Donnell[4], Peter Bock[7], Helen Ayles[2,10], Sarah Fidler[11], Richard Hayes[1], on behalf of the PopART study team¶

1 Department of Infectious Disease Epidemiology, International Statistics and Epidemiology Group, London School of Hygiene & Tropical Medicine, London, United Kingdom, 2 Zambart, School of Public Health, University of Zambia, Lusaka, Zambia, 3 Big Data Institute, Li Ka Shing Centre for Health Information and Discovery, University of Oxford, Oxford, United Kingdom, 4 Fred Hutchinson Cancer Center, Seattle, Washington, United States of America, 5 FHI 360, HIV Prevention Trials Network, Durham, North Carolina, United States of America, 6 Department of Pathology, Johns Hopkins University School of Medicine, Baltimore, Maryland, United States of America, 7 Department of Paediatrics and Child Health, Desmond Tutu TB Centre, Stellenbosch University, Cape Town, South Africa, 8 Faculty of Medicine and Health, School of Public Health, University of Sydney, Sydney, Australia, 9 Global Health and Development Department, Faculty of Public Health and Policy, LSHTM, London, United Kingdom, 10 Department of Clinical Research, Faculty of Infectious and Tropical Diseases, LSHTM, London, United Kingdom, 11 Department of Infectious Disease, Faculty of Medicine, Imperial College London, London, United Kingdom

¶ Membership of the PopART study team is listed in the Acknowledgments.
* david.macleod@lshtm.ac.uk

## Abstract

Migrants have been identified as a population left behind by the AIDS (Acquired Immune Deficiency Syndrome) response, with evidence showing poorer HIV (Human Immunodeficiency Virus) outcomes and reduced intervention effectiveness in mobile populations. We used data from the HPTN 071 (PopART) trial (ClinicalTrials.gov number, NCT01900977) to investigate migration and HIV-related indicators, assessing whether community migration influenced PopART trial results and whether migration was associated with HIV status and position on the care continuum. The PopART trial, conducted in Zambia and South Africa (SA) from November 2013 to June 2018, evaluated a universal testing and treatment intervention using a three-arm design. A cohort of 18–44-year-olds was followed annually to estimate HIV incidence, with migration out of trial communities tracked using this cohort. migration into and within the community was tracked using intervention delivery data in community members aged 18 + . HIV-related indicators were HIV status, knowledge of HIV-positive status and ART use. Migration's influence on the trial HIV incidence results was analysed using a two-stage approach for cluster-randomised trials, adjusting for community-level migration. Associations between HIV-related indicators and

**Data availability statement:** The datasets are freely available to download from the Harvard Dataverse: https://dataverse.harvard.edu/dataverse/HPTN-071.

**Funding:** This work was supported by funding awarded to Richard Hayes from the National Institute of Allergy and Infectious Diseases, the US President's Emergency Plan for AIDS Relief, the International Initiative for Impact Evaluation, the Bill and Melinda Gates Foundation, the National Institute on Drug Abuse, and the National Institute of Mental Health. The funders had no role in study design, data collection and analysis, decision to publish, or preparation of the manuscript.

**Competing interests:** The authors have declared that no competing interests exist.

both out-migration (Poisson regression using cohort data) and in-migration (logistic regression using cross-sectional data) were also estimated. While migration differed between trial arms, there was no evidence that it confounded the intervention effect on HIV incidence. There was evidence out-migration was higher among HIV-positive individuals who did not know (or did not disclose) their HIV-positive status compared to those HIV-negative (adjusted rate ratio: Zambia 1.28, 95%CI 1.17-1.39; SA 1.27, 95%CI 1.17-1.38). Residents who had moved into the community within the previous year were less likely to be aware of their HIV-positive status than longer-term residents (adjusted odds ratio: Zambia 0.18, 95%CI 0.16-0.19; SA 0.23, 95%CI 0.20-0.28) and contributed to approximately one in four of the newly identified HIV infections. Following intervention delivery the gap in knowledge of HIV status and ART treatment coverage between recent in-migrants and longer-term residents closed. Countries with high HIV burden should aim to ensure a sustained delivery of HIV services in areas with high levels of population mobility and in areas with moderate to high HIV prevalence.

## Introduction

Despite significant progress in the reduction of global HIV (Human Immunodeficiency Virus) incidence, from 2.2 million new infections per year in 2010 to 1.5 million in 2020, the UN target of fewer than 500,000 new infections per year by 2020 was not met [1,2].

In 2014 UNAIDS (The Joint United Nations Programme on HIV/AIDS) listed migrants as one of 12 populations left behind by the AIDS (Acquired Immune Deficiency Syndrome) response [3]. A systematic review in 2017 looking at the association between mobility/migration and HIV incidence in South Africa (SA) identified that those who were classified as mobile had an estimated 69% increase in the risk of acquiring HIV [4]. Migrants have also been found to be at greater risk of delays in seeking care and are also more likely to drop out at each stage of the HIV care continuum [5–9].

Universal testing and treatment (UTT) has been proposed as a potentially important tool to contribute towards HIV prevention, with four trials providing evidence that UTT can increase population-level viral suppression in sub-Saharan Africa, in turn reducing HIV incidence [10]. The HIV Prevention Trials Network (HPTN) 071 (PopART) trial, the largest of these trials, published results showing that a combination prevention package of universal testing and treatment was associated with a 20% reduction in the incidence of HIV in communities receiving the intervention compared to control communities [11].

However, the effectiveness of HIV interventions can be adversely affected by population mobility [12]. If a population is mobile then it may be difficult to achieve high population coverage of an intervention such as UTT. Also, in a trial setting where certain communities have been selected to be included in the trial, new migrants to those communities will not have received the intervention if they are from a place not

also covered by the trial. Therefore it is possible that the observed effectiveness of UTT in targeted communities, as in the PopART trial, might be reduced in the presence of high migration [3,4,7].

Data collected during the PopART trial provide an opportunity to perform a post-hoc analysis to investigate the relationship between migration and HIV-related indicators. This paper has three aims:

1) To describe the level of migration out of the trial communities and whether it is associated with an individual's HIV status or position on the care continuum

2) To investigate whether differences between trial arms in the level of community migration had any influence on the overall estimated effect of UTT on HIV incidence, the primary endpoint of the trial

3) To identify whether individuals who have recently migrated into or moved within a community differ from longer-term residents in terms of HIV prevalence, knowledge of HIV-positive status or ART coverage.

## Methods

### Overall PopART study design

PopART was a three-arm trial conducted in 12 Zambian and 9 South African urban and peri-urban communities (mean population 50,000, range 21,000–166,000) from November 2013 [11,13]. Communities within countries were matched into seven groups of three "triplets" (triplets 1–4 in Zambia and triplets 5–7 in South Africa) and then within each triplet the three communities were randomised to the three arms. Arms A and B received a combination intervention package, delivered by community HIV care providers (CHiPs), while arm C received standard care. The intervention included HIV counselling, rapid HIV testing and support for linkage to HIV care and treatment and antiretroviral therapy (ART) adherence. Additionally, from the outset of the trial, people in arm A communities diagnosed as HIV-positive were offered to start ART immediately, while in arms B and C ART was provided according to national guidelines (initially CD4 count <350 cells per μL). In 2016, national guidelines changed to immediate ART for all, which was then implemented in all three arms.

### Population cohort and out-migration

Within all three trial arms, a cohort (population cohort, PC) of 18–44 year-olds was enrolled to estimate HIV incidence (the trial's primary outcome). During enrolment, participants were asked if they intended to stay in the community for the next three years; only those that answered yes were enrolled. Recruitment began on 28th November 2013 in Zambia and 15th January 2014 in South Africa. Most were enrolled during 2014, but additional enrolment occurred between 2015 and 2017 with final enrolment in Zambia on 12th July 2017 and on 14th July 2017 in South Africa. All participants were followed up approximately annually (up to three times) before the study ended in June 2018. We refer to these data as "PC data", which are used to estimate "out-migration", defined as those who move from a study community to outside of that community.

At each annual follow-up, if an individual was not found, but investigators believed they could locate them in future, they were retained and classed as a missed visit. However, if locating them in future seemed unlikely then they were terminated from the study. Participants were also terminated from the study if they voluntarily withdrew. Reasons for termination were recorded at the time. Individuals whose reason for exiting from the study was "relocated outside of the study community" were classed as migrating out of a community. The date of out-migration was defined as the midpoint between the final attempted visit and the most recent successful visit. Individuals who had died had date of death estimated in the same manner. If an individual exited from the study for any other reason their date of exit was set as the last attempted visit date. All individuals exited the study by June 2018.

At each visit participants had a blood sample taken for laboratory testing to ascertain HIV status. Tests were performed in central laboratories in South Africa and Zambia, with additional quality control performed at the HPTN Laboratory

Center in Baltimore, USA. Participants were also asked whether they had previously had an HIV test and, if so, whether it was positive. If they answered yes to both these questions and were also confirmed as HIV-positive from the lab result they were classed as knowing their HIV-positive status. Participants who reported that they were HIV-positive were asked if they were currently on ART. This meant HIV status could be classified into four categories (after excluding missing data):

1) HIV-negative

2) HIV-positive, but did not know/disclose status

3) HIV-positive, knew and disclosed status, not on ART

4) HIV-positive, knew and disclosed status, on ART

**Intervention delivery and in-migration**

The trial delivered three annual rounds (R1, R2, R3) of the intervention package to the 14 intervention communities. Each community was divided into "zones" of 500 households, with two CHiPs assigned to each zone. Each round, CHiPs attempted to visit every household, enumerate all individuals, and obtain consent to participate in the intervention. In practice each round took longer than a year to complete. R1 ran from December 2013 to June 2015, R2 from July 2015 to August 2016 and R3 from September 2016 to December 2017 (date of first intervention Zambia: 11th December 2013, SA: 1st January 2014; date of final intervention Zambia: 22nd December 2017, SA: 6th December 2017). We will refer to the data collected as part of this intervention delivery as "CHiP data".

In R3 only, participants new to the zone were asked questions, to help ascertain if they had moved from outside the community or from another zone in the same community. We define this as their "in-migration" status, which has three categories:

1) Moved community (moved into the community since R2)

2) Moved zone (moved zones since R2, from within the community)

3) Longer-term resident (resident in the same zone in R2 and R3)

In order to allocate participants to each of these categorisations we used the following process: Within each CHiP zone of ~500 households every individual enumerated was provided with a unique ID, allowing all residents to be linked within the same zone between rounds. However, if an individual moved between zones, even within the same community, that individual's records could not be linked as they would be issued with a new zone-specific identifier.

Those who were newly enumerated in a zone and consented to participate were asked whether they had ever participated previously in the intervention. If they (i) answered "yes", or (ii) answered "no" but then specified that they were previously resident in another zone within the community, they were considered to have been resident in a different zone within the same community during the previous annual round. We will refer to this group as having "moved zone", i.e., they are participants believed to have been resident in the same community in R2 and R3, but to have moved zone within that community. If they replied "no" to previous participation and gave the reason that they had not previously participated because they had moved in from outside the community, they were classed as having "moved community" they were resident in a community in R3 but are believed to have been resident in a different community at the time of R2. Those who were resident in the same zone in R2 and R3 are classed as "continuous residents". Since in-migration status could only be properly ascertained among those who consented to participate in the intervention in R3, and not among all those enumerated, the analysis of CHiP data is limited to those who participated in R3. The proportion newly enumerated in participants and non-participants was broadly similar, suggesting that this restriction may not introduce bias.

During R3 participants were asked if they had previously had a positive HIV test. Those who reported being HIV-positive were also asked if they were currently on ART. Participants who did not report being HIV-positive were offered a rapid HIV test, with the results recorded and provided to the participant immediately.

## Statistical methods

**Out-migration (PC data).** Using the PC data, the annual rate of out-migration was estimated using Poisson regression and converted to a *risk* of out-migration over a one-year period ($risk = 1 - e^{-rate}$). Community estimates were age standardised based on the population distribution enumerated during the third round of intervention delivery. For arm C communities the age distribution was assumed to be the average of the arm A and B communities within the same triplet. Country-level estimates were presented separately for men and women and were obtained as the arithmetic mean of the age-standardised community estimates.

Differences in out-migration levels between categories of HIV status were expressed as incidence rate ratios (IRRs), derived from the Poisson regression. First unadjusted, then adjusted for community, gender and age (and the gender/age interaction), then finally adjusted additionally for socio-economic status, marital status, education, employment status, alcohol use, drug use and sexual partners in last year. The variables used were all as recorded at entry to the cohort, except age which was included as a time-varying covariate.

**Assessing the effect of out-migration on trial HIV incidence results (PC data).** Analysis to compare HIV incidence among trial arms used the same methods as in the original trial analysis, but now also adjusting for the community-level estimates of the annual risk of out-migration. The methods have been described in detail previously [12] but briefly HIV incidence, among participants HIV-negative at enrolment, was compared for arm A vs arm C and arm B vs arm C using a rate ratio, calculated using a two-stage cluster-level analysis appropriate for cluster-randomised trials with <15 clusters per trial arm [13]. The original trial results were adjusted for age, gender, their interaction, triplet and the community baseline HIV prevalence, here we will additionally adjust for community annual risk of out-migration.

**In-migration (CHiP data).** Using CHiP data, only available in arms A and B, the overall proportions new to the community in R3, since R2, were described by age, gender, HIV status, triplet and arm. The estimated probability of being new to a community within the last year was presented by community, country, gender and age. These probabilities were estimated using the following three steps: (i) estimating the log(odds) of being new to a community since R2 using logistic regression; (ii) converting the log(odds) to a probability of in-migration; (iii) accounting for the average duration between R2 and R3 being 15 months, using the equation:

$$probability_{12months} = 1 - \left( 1 - probability_{15months} \right)^{\frac{12}{15}} .$$

The association between in-migration status and HIV status was assessed using four logistic regression models, each time using in-migration status as the explanatory variable (categorised as longer-term resident, moved zone, or moved community). The four (binary) HIV outcomes assessed were:

1) HIV-positive, among all participants that have an HIV status recorded after the R3 visit (either newly tested HIV-positive with CHiPs in R3 or were already aware of their HIV-positive status)

2) Knowledge of HIV-positive status, among all those HIV-positive (i.e., the proportion of those in outcome 1 that were already aware of their HIV-positive status)

3) ART coverage, among all those who knew their HIV-positive status (i.e., the proportion of those in outcome 2 who report currently being on ART)

4) ART coverage, among all those HIV-positive (i.e., the proportion of those in outcome 1 who report currently being on ART)

Estimates were stratified by gender, country and in-migration status among R3 participants. Among participants whose HIV status was unknown to CHiPs, because they neither self-reported HIV-positive nor tested for HIV with CHiPs nor reported testing HIV-negative elsewhere in the previous 3 months, we estimated the number of HIV-positive individuals with the assumption that HIV prevalence was the same as among individuals who accepted the offering of HIV testing in R3 (stratified on community, gender, age group, and migration status).

For all regression methods applied, the assumptions underlying the regression were checked for validity. In all methods, complete case analyses were performed so no imputation of missing data was done. Stata version 18 (Statacorp) was used to perform all analyses.

### Ethical approval

Ethical approval was granted by the University of Zambia (reference number 011-11-12), Stellenbosch University (N12/11/074) and London School of Hygiene & Tropical Medicine (6326). PC participants provided informed written consent before enrolment. Intervention participants provided verbal consent to participate and written informed consent for HIV testing. Individuals who initiated ART outside of national guidelines provided written informed consent. ClinicalTrials. gov number, NCT01900977, https://clinicaltrials.gov/study/NCT01900977.

## Results

### Out-migration (PC data)

In the population cohort, 48,301 individuals aged 18–44 years were enrolled between November 2013 and July 2017. Of these 46,998 (97.3%) had at least one subsequent visit, and therefore contributed to the analysis (Zambia: 26,577; SA: 20,421). The median follow-up time was 2.5 years in Zambia and 2.8 years in SA. In both countries a high proportion of those enrolled were women (Zambia 72.7%; SA 68.9%). In Zambia 65.9% were aged 18–30, 74.0% had attended high school and 64.9% were ever married, compared with 56.8% aged 18–30 in SA, with 81.7% attending high school and 29.2% ever married (Table 1, S1 File).

The age-standardized estimated probability of out-migration from Zambian communities in a one-year period among those aged 18–44 was 10.8% (95% CI 10.4-11.1%) for men and 9.8% (9.5-10.0%) for women. In South African communities it was 11.8% (11.4-12.2%) for men and 10.3% (10.0-10.6%) for women. Most communities had between 8–14% of participants leaving each year, although triplet 7 had fewer, and within triplets out-migration was mostly similar across arms (Fig 1). Out-migration peaked in the 25–29 year age group for men and among 20–24 year old women, in both countries. There was little difference between genders in the younger age groups, but out-migration was higher in men at older ages, with a wider gap in Zambia than in South Africa (Fig 2).

At entry to the study, HIV status was missing in 2.4% of Zambian participants and 4.7% of SA participants. In Zambia, among the 25,935 participants where HIV status was known 20,555 (79.3%) tested negative, 2,252 (8.7%) identified themselves as HIV-positive and reported being on ART, 810 (3.1%) identified as HIV-positive but not on ART and 2,318 (8.9%) did not report themselves as HIV-positive but tested positive. In South Africa there were 19,457 with a known HIV status, with 15,163 (77.9%) testing negative, 1,333 (6.9%) reporting as HIV-positive and on ART, 628 (3.2%) reporting HIV-positive but not on ART and 2,333 (12.0%) testing positive but not self-reporting as HIV-positive (Table 1).

In both countries there was evidence (p<0.001) that the rate of out-migration differed by HIV status at entry to the study. In Zambia there was very little difference in the proportion migrating out of the communities among those who were HIV-negative and those who disclosed their HIV-positive status; however there was evidence that those individuals who were HIV-positive but did not know (or did not disclose) this at entry to the study were more likely to move out of their community. They were estimated to have 1.28 times the rate of out-migration compared to HIV-negative individuals (Adjusted Incidence Rate Ratio, aIRR 95% CI 1.17-1.39) (Table 2). In SA this figure was very similar (aIRR 1.27, 95% CI

PLOS Global Public Health

**Table 1. Participant characteristics among those with any follow-up information in the population cohort (N=46,998).**

| | Category | Zambia (N=26,577, Median follow up 2.5 years) | | SA (N=20,421, Median follow up 2.8 years) | |
|---|---|---|---|---|---|
| | | Distribution at entry, N (%) | Migrated out during follow up period, N | Distribution at entry, N (%) | Migrated out during follow up period, N |
| Gender | Men | 7,260 (27.3) | 2,054 | 6,321 (31.1) | 1,835 |
| | Women | 19,311 (72.7) | 4,425 | 14,002 (68.9) | 3,873 |
| Age group | 18-19 | 3,953 (14.9) | 1,152 | 1,954 (9.6) | 598 |
| | 20-24 | 7,997 (30.1) | 2,397 | 5,036 (24.8) | 1,652 |
| | 25-29 | 5,555 (20.9) | 1,418 | 4,544 (22.4) | 1,456 |
| | 30-34 | 4,122 (15.5) | 823 | 3,721 (18.3) | 963 |
| | 35-39 | 2,906 (10.9) | 427 | 2,758 (13.6) | 606 |
| | 40-44 | 2,037 (7.7) | 262 | 2,306 (11.4) | 432 |
| HIV status at entry | HIV-negative | 20,555 (79.3) | 5,119 | 15,163 (77.9) | 4,113 |
| | HIV-positive but did not know/disclose status | 2,318 (8.9) | 648 | 2,333 (12.0) | 835 |
| | HIV-positive, disclosed status, not on ART | 810 (3.1) | 180 | 628 (3.2) | 214 |
| | HIV-positive, disclosed status, on ART | 2,252 (8.7) | 365 | 1,333 (6.9) | 355 |
| Trial arm | A | 9,305 (35.0) | 2,135 | 7,021 (34.4) | 2,036 |
| | B | 7,844 (29.5) | 1,829 | 6,961 (34.1) | 1,944 |
| | C | 9,428 (35.5) | 2,521 | 6,439 (31.5) | 1,803 |
| Triplet | 1(Zambia)/ 5(SA) | 5,548 (20.9) | 1,322 | 7,098 (34.8) | 2,251 |
| | 2(Zambia)/ 6(SA) | 7,557 (28.4) | 1,603 | 7,703 (37.7) | 2,532 |
| | 3(Zambia)/ 7(SA) | 6,785 (25.5) | 1,939 | 5,620 (27.5) | 1,000 |
| | 4(Zambia) | 6,687 (25.2) | 1,621 | | |

*Missing values excluded from table. These are described in S3 File along with a more comprehensive list of sociodemographic variables.

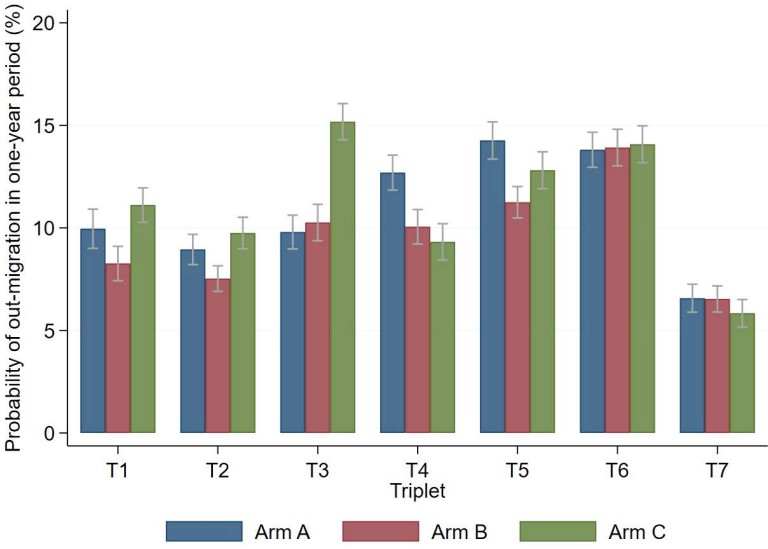

**Fig 1. Estimated probability (with 95% CI) of out-migration among 18-44 year olds in a one-year period across each of the 21 communities. [Footnote: T1= Triplet 1, T2 = triplet 2 etc.].**

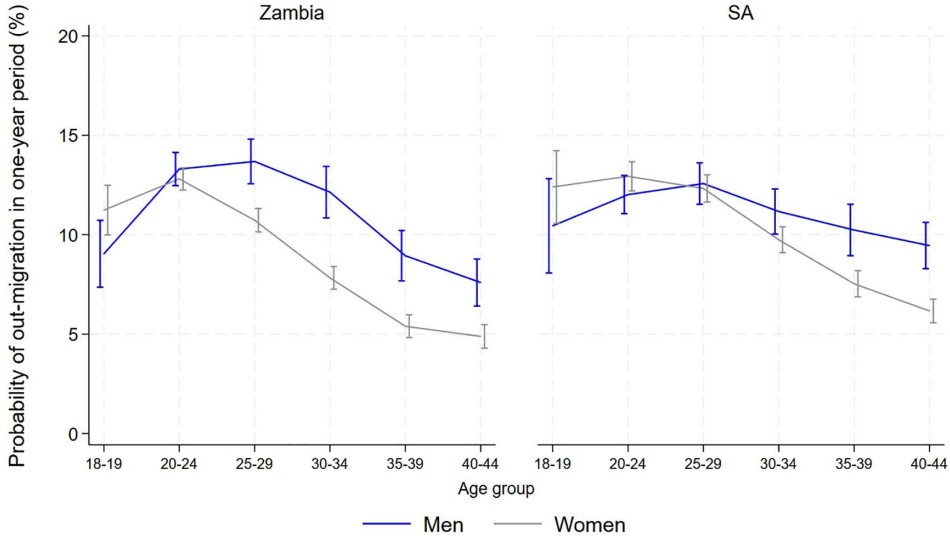

**Fig 2. Estimated probability (with 95% CI) of out-migration in the Zambian and South African communities, by age and gender.**

**Table 2. Association between HIV/ART status at entry to the study and rate of out-migration.**

| | N (%) | Out-migrated | Person years of follow up | Crude rate per 100 person years (95% CI) | Unadjusted Rate Ratio (95% CI) | Partially-adjusted Rate Ratio* (95% CI) | Fully adjusted Rate Ratio† (95% CI) |
|---|---|---|---|---|---|---|---|
| **Zambia** | | | | | | | |
| HIV-negative | 20,555 (79.3) | 5,119 | 47,000 | 10.9 (10.6-11.2) | 1 (baseline) | 1 (baseline) | 1 (baseline) |
| HIV-positive but did not know/disclose status | 2,318 (8.9) | 648 | 5,129 | 12.6 (11.7-13.6) | 1.16 (1.07-1.26) | 1.31 (1.20-1.42) | 1.28 (1.17-1.39) |
| HIV-positive, disclosed status, not on ART | 810 (3.1) | 180 | 2,107 | 8.5 (7.4-9.9) | 0.78 (0.68-0.91) | 1.03 (0.88-1.19) | 0.99 (0.85-1.16) |
| HIV-positive, disclosed status, on ART | 2,252 (8.7) | 365 | 5,237 | 7.0 (6.3-7.7) | 0.64 (0.58-0.71) | 0.94 (0.84-1.05) | 0.95 (0.85-1.07) |
| **SA** | | | | | | | |
| HIV-negative | 15,163 (77.9) | 4,113 | 38,188 | 10.7 (10.4-11.1) | 1 (baseline) | 1 (baseline) | 1 (baseline) |
| HIV-positive but did not know/disclose status | 2,333 (12.0) | 835 | 5,446 | 15.2 (14.2-16.3) | 1.42 (1.32-1.53) | 1.30 (1.21-1.41) | 1.27 (1.17-1.38) |
| HIV-positive, disclosed status, not on ART | 628 (3.2) | 214 | 1,490 | 14.4 (12.6-16.4) | 1.35 (1.18-1.55) | 1.34 (1.17-1.54) | 1.26 (1.09-1.46) |
| HIV-positive, disclosed status, on ART | 1,333 (6.9) | 355 | 3,167 | 11.2 (10.1-12.4) | 1.06 (0.95-1.18) | 1.16 (1.03-1.30) | 1.12 (1.00-1.27) |

p-value from Poisson regression for association between HIV status and out-migration was < 0.001 in all models and both countries.

[* Adjusted for community, gender, age and the age-gender interaction † Adjusted for community, gender, age, socio-economic status, marital status, education, employment status, alcohol use, drug use, sexual partners in last year and the age-gender interaction].

1.17-1.38), but in SA evidence was also found that those who knew and disclosed their HIV-positive status also had a similarly high rate of out-migration (Table 2). The association between all other variables and out-migration can be found in S1 File and S2 File.

## Assessing the effect of migration on trial result for HIV incidence (PC data)

The results from the primary analysis to compare HIV incidence between trial arms were an estimated aIRR comparing arm A to arm C of 0.88 (95% CI 0.71-1.10) and comparing B to C it was 0.68 (95% CI 0.55-0.84). After further adjusting for community-level estimates of migration there was little change in these effects, with an aIRR of 0.89 (0.70-1.13) comparing A to C and 0.73 (0.58-0.93) comparing arm B to C (further details on this analysis are provided in S3 File).

## In-migration (CHiP data)

Among the 181,418 Zambian participants aged ≥18 years in R3 of the intervention delivery, 107,839 (59.4%) were women and 137,078 (75.6%) were aged 18–40. Overall 22,495 (12.4%) had moved into the community since R2 with a further 31,925 (17.6%) who were classified as having moved into the *zone* since R2 (but from within the same community). In the 78,947 SA participants 46,493 (58.9%) were women, with 54,594 (69.1%) aged 18–40, with 8,109 (10.3%) moving in from outside the community and 12,766 (16.2%) moving into the zone from within the community since the previous round. (Table 3) Fig 3 shows the estimated proportion of inhabitants new to each of the 14 communities within the previous year, with Fig 4 showing the estimates within each country by age and gender.

There was little difference in the prevalence of HIV among the three categories of in-migration in either country (Table 4). However, strong evidence was found of a difference in knowledge of HIV-positive status (prior to the R3 CHiP offer of HIV testing) across the categories of movement among people living with HIV (PLHIV). In Zambia, 89.7% of HIV-positive longer-term residents knew their status, compared to 74.5% of those new to the zone and 58.8% of those new to the community. Adjusting for age and gender this suggested that PLHIV new to the community had less than a fifth of the odds of knowing their HIV-positive status compared to longer-term residents (Table 4). In SA a similar pattern was observed, with 92.3% of longer-term residents aware of their HIV-positive status, versus 82.0% of those new to the zone and 72.1% of those new to the community (Table 4). Thus, during R3, of the 5,598 positive HIV test results across both countries 3,006 (53.7%) were from individuals new to the zone or community, despite comprising only 28.9% of the participants.

Although in SA there was no evidence of a difference in the proportion on ART across the movement groups (among those who knew their HIV-positive status), in Zambia there was strong evidence (p<0.0001) that HIV-positive participants who knew and disclosed their HIV-positive status were *more* likely to be on ART if they were new to the community than if they were longer-term residents. However, the differences were small in absolute terms: 90.0% of longer-term residents who reported being HIV-positive reported being on ART, compared with 92.9% and 93.4% in those new to the zone and community respectively (Table 4).

Combining these two measures to give the proportion on ART among all known HIV-positive participants still showed that PLHIV in Zambia who were new to the community had about a third of the odds of being on ART compared to longer-term residents (about two-fifths of the odds in SA). Among all known HIV-positive participants, in Zambia 80.7% of longer-term residents were on ART compared with 54.9% of those who were new to the community and in South Africa 81.7% of longer-term residents were on ART compared with 61.4% of those who were new to the community (Table 4).

This gap between the three movement groups was reduced by the intervention; Fig 5 shows the proportion of known HIV-positive participants who were on ART, the proportion that knew their status but were not on ART and the proportion that did not know (or disclose) their HIV-positive status prior to the offer of HIV testing in R3, stratified by gender and country. Within each panel the bars on the left show the proportions just prior to the delivery of the R3 CHiP testing, and the bars on the right show the estimates at the end of the round (R3) by which time all offers of testing had been completed in all communities and those testing positive had time to link to care. This illustrates that ART coverage increased in all three movement groups, and that the gap between the three categories was reduced, following delivery of the intervention in R3. Further detail on the above is available in S4 File.

**Table 3. Participant characteristics of CHiP data at round 3.**

| | | R3 Participants N (%*) | Longer-term resident N(%†) | New to the zone since R2 (but not the community) N(%†) | New to the community since R2 N (%†) |
|---|---|---|---|---|---|
| **ZAMBIA** | | | | | |
| All | All | 181,418 | 126,998 (70.0) | 31,925 (17.6) | 22,495 (12.4) |
| Gender | Men | 73,579 (40.6) | 53,083 (72.1) | 11,687 (15.9) | 8,809 (12.0) |
| | Women | 107,839 (59.4) | 73,915 (68.5) | 20,238 (18.8) | 13,686 (12.7) |
| Age group | 18-19 | 16,837 (9.3) | 11,295 (67.1) | 2,801 (16.6) | 2,741 (16.3) |
| | 20-24 | 43,515 (24.0) | 27,052 (62.2) | 8,937 (20.5) | 7,526 (17.3) |
| | 25-29 | 32,622 (18.0) | 20,358 (62.4) | 7,260 (22.3) | 5,004 (15.3) |
| | 30-34 | 25,050 (13.8) | 17,108 (68.3) | 5,038 (20.1) | 2,904 (11.6) |
| | 35-39 | 19,054 (10.5) | 13,936 (73.1) | 3,353 (17.6) | 1,765 (9.3) |
| | 40-44 | 13,754 (7.6) | 10,841 (78.8) | 1,877 (13.6) | 1,036 (7.5) |
| | 45-49 | 8,878 (4.9) | 7,276 (82.0) | 1,038 (11.7) | 564 (6.4) |
| | 50-54 | 6,475 (3.6) | 5,526 (85.3) | 632 (9.8) | 317 (4.9) |
| | 55-59 | 4,725 (2.6) | 4,194 (88.8) | 340 (7.2) | 191 (4.0) |
| | 60-64 | 3,668 (2.0) | 3,262 (88.9) | 250 (6.8) | 156 (4.3) |
| | 65+ | 6,840 (3.8) | 6,150 (89.9) | 399 (5.8) | 291 (4.3) |
| HIV status after R3 visit | Tested HIV-negative | 119,709 (66.0) | 82,271 (68.7) | 21,136 (17.7) | 16,302 (13.6) |
| | Tested HIV-positive | 4,381 (2.4) | 1,962 (44.8) | 1,238 (28.3) | 1,181 (27.0) |
| | Self-reported HIV-positive, not on ART | 2,078 (1.1) | 1,708 (82.2) | 258 (12.4) | 112 (5.4) |
| | Self-reported HIV-positive, on ART | 20,229 (11.2) | 15,299 (75.6) | 3,356 (16.6) | 1,574 (7.8) |
| | Did not test | 35,021 (19.3) | 25,758 (73.6) | 5,937 (17.0) | 3,326 (9.5) |
| Trial arm | A | 90,425 (49.8) | 64,946 (71.8) | 14,538 (16.1) | 10,941 (12.1) |
| | B | 90,993 (50.2) | 62,052 (68.2) | 17,387 (19.1) | 11,554 (12.7) |
| Triplet | 1 | 31,512 (17.4) | 23,675 (75.1) | 3,593 (11.4) | 4,244 (13.5) |
| | 2 | 36,212 (20.0) | 26,054 (71.9) | 6,937 (19.2) | 3,221 (8.9) |
| | 3 | 85,316 (47.0) | 57,568 (67.5) | 16,244 (19.0) | 11,504 (13.5) |
| | 4 | 28,378 (15.6) | 19,701 (69.4) | 5,151 (18.2) | 3,526 (12.4) |
| **South Africa** | | | | | |
| All | All | 78,947 | 58,072 (73.6) | 12,766 (16.2) | 8,109 (10.3) |
| Gender | Men | 32,454 (41.1) | 23,029 (71.0) | 5,592 (17.2) | 3,833 (11.8) |
| | Women | 46,493 (58.9) | 35,043 (75.4) | 7,174 (15.4) | 4,276 (9.2) |
| Age group | 18-19 | 4,681 (5.9) | 3,365 (71.9) | 725 (15.5) | 591 (12.6) |
| | 20-24 | 14,443 (18.3) | 9,212 (63.8) | 2,857 (19.8) | 2,374 (16.4) |
| | 25-29 | 13,930 (17.6) | 9,130 (65.5) | 2,828 (20.3) | 1,972 (14.2) |
| | 30-34 | 12,477 (15.8) | 8,859 (71.0) | 2,350 (18.8) | 1,268 (10.2) |
| | 35-39 | 9,063 (11.5) | 6,874 (75.8) | 1,460 (16.1) | 729 (8.0) |
| | 40-44 | 7,255 (9.2) | 5,888 (81.2) | 922 (12.7) | 445 (6.1) |
| | 45-49 | 5,415 (6.9) | 4,556 (84.1) | 606 (11.2) | 253 (4.7) |
| | 50-54 | 4,423 (5.6) | 3,875 (87.6) | 375 (8.5) | 173 (3.9) |
| | 55-59 | 2,993 (3.8) | 2,619 (87.5) | 253 (8.5) | 121 (4.0) |
| | 60-64 | 1,945 (2.5) | 1,691 (86.9) | 204 (10.5) | 50 (2.6) |
| | 65+ | 2,322 (2.9) | 2,003 (86.3) | 186 (8.0) | 133 (5.7) |

*(Continued)*

**Table 3.** (Continued)

| | | R3 Participants N (%*) | Longer-term resident N(%†) | New to the zone since R2 (but not the community) N(%†) | New to the community since R2 N (%†) |
|---|---|---|---|---|---|
| HIV status after R3 visit | Tested HIV-negative | 43,249 (54.8) | 31,393 (72.6) | 7,017 (16.2) | 4,839 (11.2) |
| | Tested HIV-positive | 1,217 (1.5) | 630 (51.8) | 307 (25.2) | 280 (23.0) |
| | Self-reported HIV-positive, not on ART | 1,154 (1.5) | 871 (75.5) | 176 (15.3) | 107 (9.3) |
| | Self-reported HIV-positive, on ART | 8,532 (10.8) | 6,698 (78.5) | 1,218 (14.3) | 616 (7.2) |
| | Did not test | 24,795 (31.4) | 18,480 (74.5) | 4,048 (16.3) | 2,267 (9.1) |
| Trial arm | A | 39,694 (50.3) | 28,363 (71.5) | 6,720 (16.9) | 4,611 (11.6) |
| | B | 39,253 (49.7) | 29,709 (75.7) | 6,046 (15.4) | 3,498 (8.9) |
| Triplet | 5 | 21,859 (27.7) | 16,382 (74.9) | 3,452 (15.8) | 2,025 (9.3) |
| | 6 | 41,228 (52.2) | 29,016 (70.4) | 7,338 (17.8) | 4,874 (11.8) |
| | 7 | 15,860 (20.1) | 12,674 (79.9) | 1,976 (12.5) | 1,210 (7.6) |

\* Column percentages (i.e., the proportion in each characteristic category out of the total number of participants).

† Row percentages (i.e., the proportion of the people in each in-migration category out of the participants in that category of characteristic).

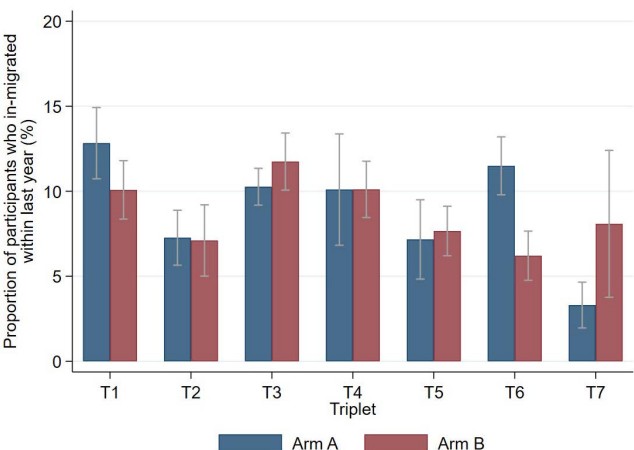

**Fig 3. Estimated probability (with 95% CI) of being a recent in-migrant (moved into a community within the last 12 months) across each of the 14 arm A and B communities during final intervention round. [Footnote: T1= Triplet 1, T2 = triplet 2 etc.].**

## Discussion

Our analysis indicates a strong relationship between migration and HIV-related indicators. We estimated about 10% of community members aged 18–44 left the trial communities each year, and the proportions were fairly consistent across Zambian and SA communities that ranged in size from 21,000–166,000. We found that individuals who were HIV-positive but not aware of their status when they entered the study were more likely to migrate out of a community than those on ART or who were HIV-negative. This association persisted after controlling for age, gender, number of sexual partners, and marital status. These results are primarily applicable to long-term migration (rather than short-term, seasonal migration) as participants who temporarily relocated should have been retained in the cohort.

Previous studies have reported higher HIV prevalence among in-migrants compared with longer-term residents in contrast to our observed result of similar HIV prevalence among residents aged 18 who were recent in-migrants and those

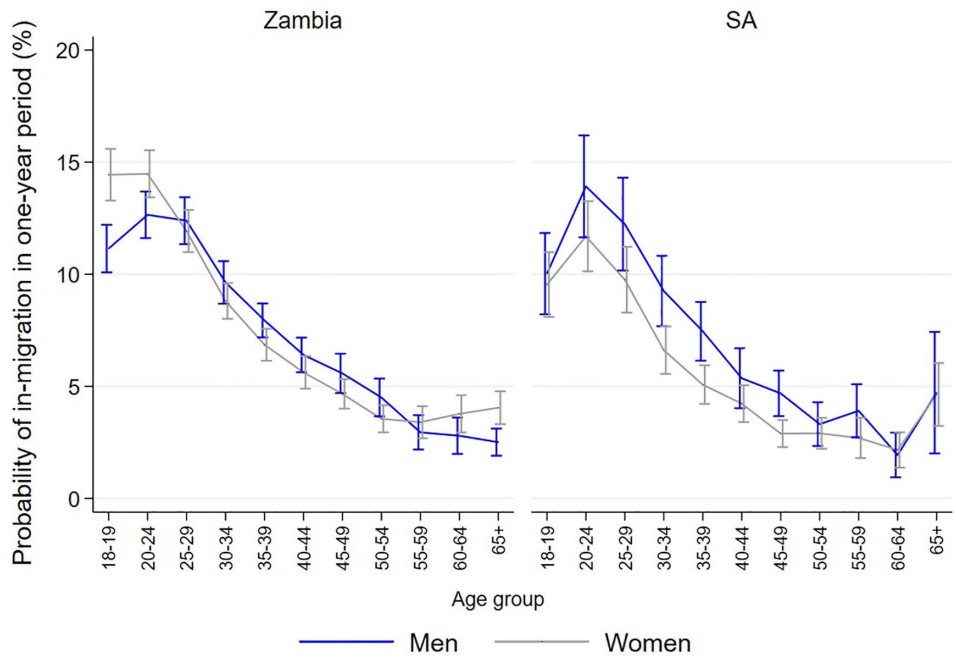

**Fig 4. Estimated probability (with 95% CI) of being a recent in-migrant (moved into a community within the last 12 months) in the Zambian and South African arm A and B communities, by age and gender during final intervention round.**

who had lived in the community longer. For example, in Rakai, Uganda, Grabowski et al. found that people moving into high-prevalence "hotspot" communities had a higher HIV prevalence than those who had lived there longer, particularly among women in agrarian communities [14]. Lurie et al. observed in KwaZulu-Natal that community members who had migrated within previous two years had a greater prevalence of HIV, and similarly Anglewicz et al. found an increased odds of migration among HIV-positive men and women in Malawi [15,16]. The differences between these studies and ours may be due to the different epidemic contexts, differing definitions of migration but also the influence of the widespread testing and treatment occurring within the PopART study.

However, HIV-positive in-migrants were less likely to know their HIV-positive status than longer-term residents, which is likely explained by their lower likelihood of having been previously exposed to the UTT intervention. However, those who moved zone within the community (so should have received the intervention) also had lower knowledge of their status than longer-term residents, while still higher than in-migrants. This may be because more mobile individuals are harder to reach with the intervention or are representative of groups less likely to test. In-migrants may have felt less comfortable disclosing their HIV-positive status to CHiPs in R3 as they were yet to build a relationship with them. In R3 those who had moved into or within the community contributed 54% of the newly identified HIV-positive cases, but comprised only 29% of the population. This emphasises the importance of repeated UTT interventions in communities with high mobility. In-migrants who knew/disclosed their status had comparable or better ART coverage than longer-term residents.

We did not observe large differences in the results observed in Zambia and SA. Overall levels of out-migration from the communities were similar in the two countries, and the HIV indicator most strongly associated with the rate of out-migration in both countries was not being aware of HIV-positive status at study entry, with a similar magnitude of estimated effect. In Zambia the estimated rate out-migration among those who did know their HIV-positive status was similar to HIV-negative individuals but in SA there was some evidence that this group were more likely to migrate. Our observed associations of HIV-related indicators and in-migration were also similar between countries, with one exception being in

**Table 4. Association between in migration category and HIV status at R3 visit. Across the three categories of in-migration status we compare: (a) The proportion of participants that are known to be HIV-positive among all participants that have an HIV status recorded after R3 visit. (b) The proportion of participants that knew (and disclosed) their HIV-positive status, among all participants known to be HIV-positive after R3 visit. (c) The proportion of participants who are on ART among PLHIV who knew (and disclosed) their HIV-positive status at the R3 visit. (d) The proportion of participants who are on ART among all PLHIV.**

| Migration category | ZAMBIA | | | | SA | | | |
|---|---|---|---|---|---|---|---|---|
| | n/N (%) | Unadjusted OR | Adjusted OR* | p-value† | n/N (%) | Unadjusted OR | Adjusted OR* | p-value† |
| **(a)HIV-positive among those where HIV status is known after R3 visit** | | | | | | | | |
| Longer-term resident | 18,969/101,240 (18.7%) | Reference | Reference | 0.0092 | 8,199/39,592 (20.7%) | Reference | Reference | 0.0651 |
| Moved zone | 4,852/25,988 (18.7%) | 0.93 (0.89-0.96) | 1.06 (1.02-1.10) | | 1,701/8,718 (19.5%) | 0.81 (0.76-0.86) | 0.96 (0.90-1.02) | |
| Moved community | 2,867/19,169 (15.0%) | 0.73 (0.70-0.76) | 1.00 (0.96-1.05) | | 1,003/5,842 (17.2%) | 0.66 (0.61-0.71) | 0.92 (0.85-0.99) | |
| **(b)Knew status (prior to R3 visit) among those known HIV-positive after R3 visit** | | | | | | | | |
| Longer-term resident | 17,007/18,969 (89.7%) | Reference | Reference | <0.0001 | 7,569/8,199 (92.3%) | Reference | Reference | <0.0001 |
| Moved zone | 3,614/4,852 (74.5%) | 0.34 (0.31-0.37) | 0.36 (0.33-0.40) | | 1,394/1,701 (82.0%) | 0.37 (0.32-0.43) | 0.42 (0.35-0.49) | |
| Moved community | 1,686/2,867 (58.8%) | 0.15 (0.14-0.17) | 0.18 (0.16-0.19) | | 723/1,003 (72.1%) | 0.20 (0.17-0.24) | 0.23 (0.20-0.28) | |
| **(c)On ART among those who knew their HIV-positive status prior to R3** | | | | | | | | |
| Longer-term resident | 15,299/17,007 (90.0%) | Reference | Reference | <0.0001 | 6,698/7,569 (88.5%) | Reference | Reference | 0.2071 |
| Moved zone | 3,356/3,614 (92.9%) | 1.47 (1.28-1.69) | 1.64 (1.43-1.89) | | 1,218/1,394 (87.4%) | 0.91 (0.76-1.08) | 1.00 (0.83-1.20) | |
| Moved community | 1,574/1,686 (93.4%) | 1.57 (1.29-1.92) | 1.79 (1.46-2.19) | | 616/723 (85.2%) | 0.72 (0.58-0.90) | 0.81 (0.65-1.02) | |
| **(d)On ART among all those known HIV-positive after R3 visit** | | | | | | | | |
| Longer-term resident | 15,299/18,969 (80.7%) | Reference | Reference | <0.0001 | 6,698/8,199 (81.7%) | Reference | Reference | <0.0001 |
| Moved zone | 3,356/4,852 (69.2%) | 0.54 (0.50-0.58) | 0.61 (0.56-0.66) | | 1,218/1,701 (71.6%) | 0.56 (0.49-0.63) | 0.64 (0.56-0.73) | |
| Moved community | 1,574/2,867 (54.9%) | 0.28 (0.26-0.31) | 0.33 (0.30-0.36) | | 616/1,003 (61.4%) | 0.33 (0.29-0.39) | 0.40 (0.34-0.46) | |

p-values obtained by logistic regression.

[* Adjusted for community, gender, age and the age-gender interaction † p-value is from adjusted analysis].

Zambia we observed that ART coverage among those known to be HIV-positive was higher among in-migrants than in longer-term residents whereas in SA there was some weak evidence ART coverage was lower in in-migrants who already were aware of their status.

The PopART trial analysis found that arm A had a higher HIV incidence than arm B, despite arm A introducing the policy of immediate ART initiation regardless of CD4 count sooner than in arm B [11]. We observed that arm A communities had higher levels of migration than arm B, but adjusting the trial results for community-level estimates of migration did not change the estimates of the effect of trial arm on HIV incidence. Despite community-level migration being associated with HIV incidence, it seems that matching of triplets and adjustment for age, gender and baseline HIV prevalence already accounted indirectly for the confounding effect of migration. Therefore, we found no evidence to suggest that differing migration across trial arms explained differences in HIV incidence between arms A and B.

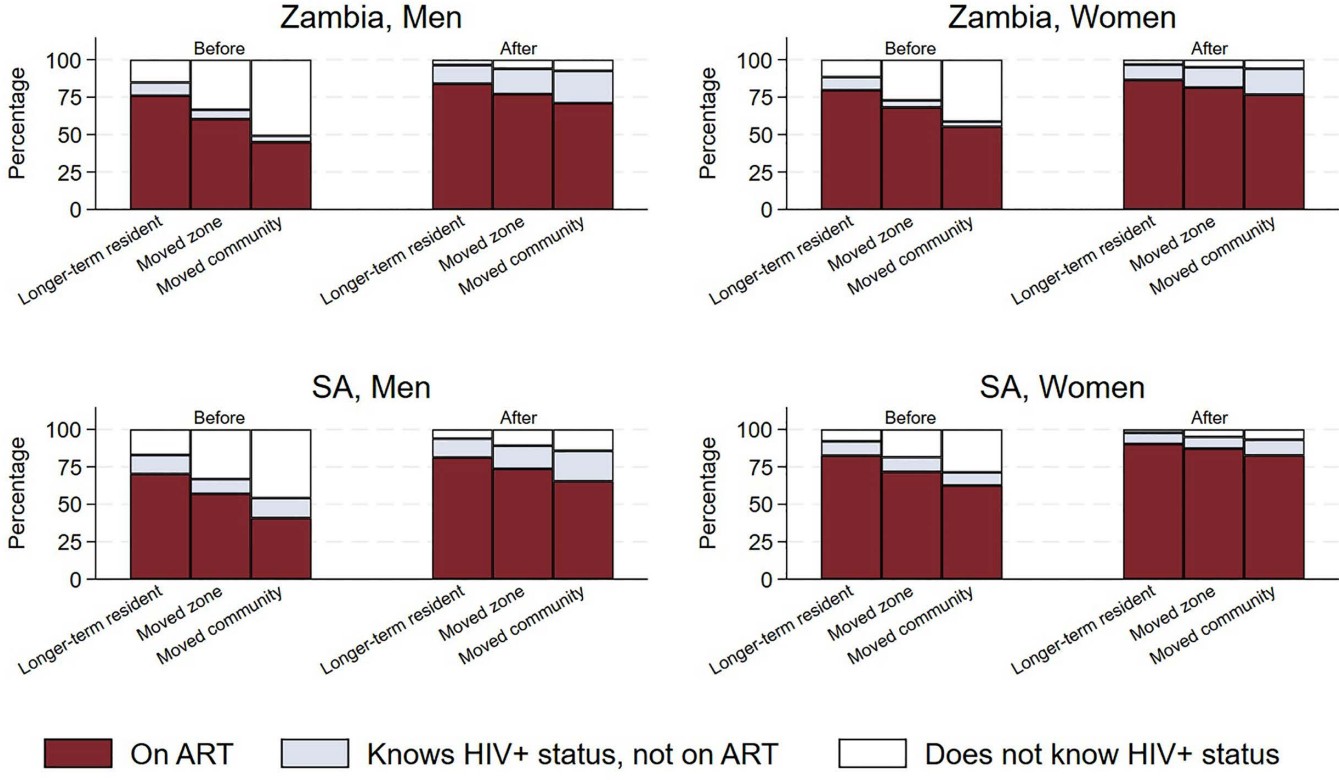

**Fig 5. Proportion on ART, proportion that know HIV-positive status but not on ART and proportion that do not know HIV-positive status** *among all HIV-positive participants*, **prior to and at the end of the PopART R3 intervention.** HIV prevalence among those who did not self-report being HIV-positive was assumed to be the same among individuals who accepted the offer of testing and those who did not (stratified by age group, gender, community and in-migration category).

The ANRS 12249 UTT trial investigators also examined the potential influence of migration on their results and concluded that migration did have an influence, with the flow of new infections into the communities attenuating the effect of their intervention on population-level ART coverage and viral suppression [7].

Our study has limitations. PC participants were only included if they intended to stay in the community for three years, potentially underestimating out-migration. We could not reliably ascertain whether HIV status changed before out-migration. ART use relied on self-report, risking misclassification; viral load suppression, a more objective measure, was only available at one time point (PC24) in the PC and not at all in the CHiP data. Non-disclosure of HIV-positive status could be due to an individual being unaware of their HIV status, or they were uncomfortable disclosing their status. With the CHiP data, this might differ between in-migrants and longer-term residents, due to rapport with CHiPs developed over the course of the study. Migration estimates from the CHiP data are built on some assumptions as the questions asked were used to help deliver the intervention, rather than to measure migration; therefore there may be some misclassification in categorisation, particularly in the "moved zone" (within a community) group. Finally, as the CHiP analysis could only be performed in the two intervention arms, the comparison was between in-migrants with no previous exposure to the intervention and longer-term residents with one or two years of UTT intervention; it would have been beneficial to have data from the control arm as well.

## Conclusions

PLHIV who migrated into the UTT trial communities or moved within those communities were less likely to be aware of their HIV-positive status, contributing to over half of the individuals who were newly identified as HIV-positive during R3 of the intervention. Countries with high HIV burden should consider how best to identify and engage with migrants and should ensure sustained delivery of HIV services in areas with high levels of population mobility, in order to reach and engage with mobile individuals.

## Supporting information

**S1 File. Population cohort characteristics and rate of out-migration.**
(DOCX)

**S2 File. Stratum-specific rate ratios for association between gender/age and out-migration.**
(DOCX)

**S3 File. Detail on adjustment for primary trial outcome.**
(DOCX)

**S4 File. 90–90 estimates before and after R3 stratified by migration status.**
(DOCX)

**S5 File. Members of the HPTN 071 (PopART) trial study team.**
(DOCX)

## Acknowledgments

The HPTN 071 (PopART) trial was supported by the National Institute of Allergy and Infectious Diseases, the US President's Emergency Plan for AIDS Relief, the International Initiative for Impact Evaluation, the Bill and Melinda Gates Foundation, the National Institute on Drug Abuse, and the National Institute of Mental Health. The content herein is solely the responsibility of the authors and does not necessarily represent the official views of the funding agencies.

We would like to acknowledge the contribution of the community members in the 21 communities and, in particular, the participants of the population cohort who gave their time and samples for this research. We also thank our partners in South Africa, including PEPFAR partners (Kheth'Impilo, ANOVA Healthcare, and the South African Clothing and Textile Workers Union Worker Health Program) and city of Cape Town and Western Cape Government Department of Health colleagues who have worked to implement the trial activities, and partners in Zambia, including the Zambian Ministry of Health, the Center for Infectious Disease Research in Zambia, Zambia Prevention, Care, and Treatment Partnership Project II, and JSI (John Snow Inc); the administrative and support teams at the institutions involved in this trial and the hundreds of field staff who delivered the intervention and collected the research data and the community advisory boards, in-country trial steering and management committees, international advisory group, and data and safety monitoring board for their oversight and consultation during the conduct of the trial.

## Author contributions

**Conceptualization:** David Macleod, Sian Floyd, Kwame Shanaube, William Probert.

**Data curation:** David Macleod, Justin Bwalya, Ab Schaap, Timothy Skalland, Estelle Piwowar-Manning, Nomtha Mandla, Deborah Donnell.

**Formal analysis:** David Macleod, Ab Schaap, Timothy Skalland, Deborah Donnell.

**Funding acquisition:** Sarah Fidler, Richard Hayes.

**Investigation:** Sian Floyd, Estelle Piwowar-Manning, Musonda Simwinga.

**Methodology:** David Macleod, Sian Floyd, William Probert, Richard Hayes.

**Project administration:** Ayana Moore.

**Supervision:** Sian Floyd, Richard Hayes.

**Visualization:** David Macleod.

**Writing – original draft:** David Macleod.

**Writing – review & editing:** David Macleod, Sian Floyd, Kwame Shanaube, William Probert, Justin Bwalya, Ab Schaap, Timothy Skalland, Ayana Moore, Estelle Piwowar-Manning, Graeme Hoddinott, Virginia Bond, Musonda Simwinga, Nomtha Mandla, Deborah Donnell, Peter Bock, Helen Ayles, Sarah Fidler, Richard Hayes.

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
