## [Decision Letter · Decision Letter 0]

12 Nov 2025

PGPH-D-25-02664

Migration and its impact on universal testing and treatment in the HPTN 071 (PopART) study communities

Dear Dr. Macleod,

Thank you for submitting your manuscript to PLOS Global Public Health. After careful consideration, we feel that it has merit but does not fully meet PLOS Global Public Health’s publication criteria as it currently stands. Therefore, we invite you to submit a revised version of the manuscript that addresses the points raised during the review process.

Editor comments:

The reviewers and I found the manuscript to provide useful contributions to the literature on migration and the HIV continuum of prevention and care, particularly in the context of large trials of universal test and treat. However, there are several areas that need to be clarified and strengthened.All three of the reviewers request clarifications concerning methods, and I agree with the suggestions to shift some of the supplemental methods information into the main text.I also agree with the reviewers' requests for an expanded discussion that clarifies and discusses the implications of the types of migration examined in the manuscript. Better linkage to the literature overall is needed in the discussion, and I would recommend additional attention to regional differences if any, and implications/recommendations for HIV prevention and care based on the findings.

We look forward to receiving your revised manuscript.

Kind regards,

Marie A. Brault, PhD

Academic Editor

Journal Requirements:

2. Please provide a detailed online Financial Disclosure statement. This is published with the article. It must therefore be completed in full sentences and contain the exact wording you wish to be published.

a) State the initials, alongside each funding source, of each author to receive each grant. For example: “This work was supported by the National Institutes of Health (####### to AM; ###### to CJ) and the National Science Foundation (###### to AM).”

For more information, please go to our submission guidelines:

https://journals.plos.org/globalpublichealth/s/submission-guidelines#loc-financial-disclosure-statement

3. Please ensure that the funders and grant numbers match between the Financial Disclosure field and the Funding Information tab in your submission form. Note that the funders must be provided in the same order in both places as well.

4. Please update your online Competing Interests statement. If you have no competing interests to declare, please state: “The authors have declared that no competing interests exist.”

5. In the online submission form, you indicated that “Requests for the data may be sent to HPTN-Data-Access@scharp.org.”.

a) In a public repository,

b) Within the manuscript itself, or

c) Uploaded as supplementary information.

6. Please provide separate main figure files in .tif or .eps format only and ensure that all files are under our size limit of 10MB.

Additional Editor Comments (if provided):

Reviewers' comments:

Reviewer's Responses to Questions

**Comments to the Author**

1. Does this manuscript meet PLOS Global Public Health’s publication criteria? Is the manuscript technically sound, and do the data support the conclusions? The manuscript must describe methodologically and ethically rigorous research with conclusions that are appropriately drawn based on the data presented.

Reviewer #1: Yes

Reviewer #2: Yes

Reviewer #3: Yes

2. Has the statistical analysis been performed appropriately and rigorously?

Reviewer #1: Yes

Reviewer #2: Yes

Reviewer #3: Yes

3. Have the authors made all data underlying the findings in their manuscript fully available (please refer to the Data Availability Statement at the start of the manuscript PDF file)?

Reviewer #1: No

Reviewer #2: Yes

Reviewer #3: Yes

4. Is the manuscript presented in an intelligible fashion and written in standard English?

Reviewer #1: Yes

Reviewer #2: Yes

Reviewer #3: Yes

5. Review Comments to the Author

Reviewer #1: This is an interesting study with comprehensive analyses. Nonetheless, the manuscript could be further improved in terms of clarity.

Line 158 – 162: The sentence could be improved. e.g. Using the PC data, the annual rate of out-migration was estimated using Poisson regression and converted to the risk of out-migration over a one-year period. Community estimates were age-standardised based on the population distribution enumerated in round 3 of the intervention delivery. Differences in out-migration levels between categories of HIV status were expressed as relative risks derived from the Poisson regression.

Line 164-166: The statistical test is to be clearly stated e.g. two-stage cluster-level analysis.

A statement confirming that the assumptions of the Poisson regression have been met is to be included.

Line 175-183: The type of data for the outcome variable is to be stated.

Line 226: The term adjusted rate ratio may be abbreviated as aIRR in subsequent sections, and rate ratio may likewise be abbreviated as IRR.

Ensure all percentage figures decimal points in the text are standardised e.g. 1 decimal point.

Line 276: figure 3 (cap F).

Table 1, Table 3. Supplementary Table 3: Since the symbol (%) is already indicated at the top of the table, it can be omitted from the individual values.

Table 2:- association : Cap A.

Table 2: On the first row, column 4-7, 95% CI is to be stated.

The additional statistical analyses info in the supplementary file could be moved to the Statistical Methods section.

For Table 2 footnote [* Adjusted for gender, age (and their interaction) and community † Adjusted for gender, age (and their interaction) community, socio-economic status, marital status, education, employment status, alcohol use, drug use, sexual partners in last year], the footnotes should follow those indicated in the supplementary files where applicable, and should specify the actual interaction variable(s). The same applies to Table 4. For [† Adjusted for gender, age (and their interaction) community], a comma should be placed after the closing bracket.

Supplementary material 3 table footnote: For [††Only adjusted for triplet, not community] & Line 92-93: Specify the categories (coding) for the communities in the statistical methods section.

In the tables where statistical tests were performed, the name of the test is to be indicated in the table footnote. For the statistical tests mentioned in the Statistical Methods section, these should also be clearly identified in the Results section, including in the relevant tables and figures.

The name of the statistical software, including the publisher, version, and the level of statistical significance adopted in the analyses, are to be stated.

The information on missing values and their handling in the analyses is be described in the statistical methods section.

Reviewer #2: This is a very interesting manuscript looking at important questions around migration, HIV status and treatment status, with clear and rigorous analysis underlying the findings. I have three minor requests for the authors:

1) I would ask that if possible the authors add some clarification in the discussion about the type of migration examined in this study (ie whether short-term or long-term migration) - these results seem to me more applicable to long-term migration, although some study sites are well-known for seasonal migration (e.g. between Eastern and Western Cape in South Africa).

2) Can the authors just contextualise their discussion that the results are for adults aged 18-44 (plus any additional limitations that would prevent individuals from being seen in the PC/CHIP data).

3) Can the authors confirm if the analysis is a complete-case analysis (or how missing data was dealt with).

Reviewer #3: The authors provide relevant contributions regarding HIV vulnerability among migrant populations in communities in Zambia and South Africa. The article presents a robust methodology and solid results; however, the authors need to improve the presentation of the text to improve its structure and facilitate comprehension by readers.

Title: As this is not a journal dedicated specifically to HIV/AIDS, I suggest that the authors indicate that the universal testing and treatment approach is for HIV.

Abstract: Similarly to the title, acronyms related to HIV/AIDS are not trivial for readers of this review. I suggest that authors pay attention to the acronyms used in the abstract and throughout the text for an initial explanation (e.g., HIV, AIDS, UN, UNAIDS, HPTN, etc.).

I suggest that the authors provide more detail on the HIV-related outcomes and the migration variable as the main independent variable in the analysis in the methods section of the abstract.

The authors present the result as follows: "out-migration was higher among HIV-positive individuals...". This interpretation may mislead readers into thinking that migration is the dependent variable, and HIV is the independent variable. I suggest inverting this interpretation, since the outcome is HIV (e.g., individuals who migrated out were more likely to be unaware or not disclose their HIV-positive status...).

In the results section, the authors present the associations without mentioning the measures of association and their respective confidence intervals.

Introduction: As in the abstract, I suggest that authors look into explaining the acronyms.

Between lines 79 and 86, the difference between objective 1 and objective 3 is unclear. Wouldn't objective 1 be included in objective 3?

Methods: The definition of variables and statistical analysis described in the methods are unclear. Reading the supplementary information S1 and S2, the authors should consider including them in the text of the article, as they are essential for understanding the strategies used and for the strength of the article.

Results: I suggest that the authors describe sociodemographic characteristics so that readers can better understand the study population.

Between lines 220 and 224, the situation is the same as mentioned in the abstract; I suggest inverting the interpretation to better clarify migration as an independent variable.

Discussion: I missed the authors discussing possible regional differences in the results presented. I also suggest a discussion of the results with other results in the literature.

In the first paragraph of the discussion, I also suggest that authors consider the inversion of interpretation.

Figure 1: A footnote explaining what T1, T2, T3, etc. is missing.

Figure 3: Would it be possible to increase the size of the graphs? It is not easy to compare them at their current size.

6. PLOS authors have the option to publish the peer review history of their article (what does this mean?). If published, this will include your full peer review and any attached files.

**Do you want your identity to be public for this peer review?** For information about this choice, including consent withdrawal, please see our Privacy Policy.

Reviewer #1: No

Reviewer #2: No

Reviewer #3: No

 Figure Resubmissions:

---

## [Decision Letter · Decision Letter 1]

26 Jan 2026

Migration and its impact on universal HIV testing and treatment in the HPTN 071 (PopART) study communities

PGPH-D-25-02664R1

Dear Mr Macleod,

We are pleased to inform you that your manuscript 'Migration and its impact on universal HIV testing and treatment in the HPTN 071 (PopART) study communities' has been provisionally accepted for publication in PLOS Global Public Health.

Best regards,

Marie A. Brault, PhD

Academic Editor

Reviewer Comments (if any, and for reference):

Reviewer's Responses to Questions

**Comments to the Author**

1. If the authors have adequately addressed your comments raised in a previous round of review and you feel that this manuscript is now acceptable for publication, you may indicate that here to bypass the “Comments to the Author” section, enter your conflict of interest statement in the “Confidential to Editor” section, and submit your "Accept" recommendation.

Reviewer #1: All comments have been addressed

Reviewer #2: All comments have been addressed

2. Does this manuscript meet PLOS Global Public Health’s publication criteria? Is the manuscript technically sound, and do the data support the conclusions? The manuscript must describe methodologically and ethically rigorous research with conclusions that are appropriately drawn based on the data presented.

Reviewer #1: Yes

Reviewer #2: Yes

3. Has the statistical analysis been performed appropriately and rigorously?

Reviewer #1: Yes

Reviewer #2: Yes

4. Have the authors made all data underlying the findings in their manuscript fully available (please refer to the Data Availability Statement at the start of the manuscript PDF file)?

Reviewer #1: Yes

Reviewer #2: Yes

5. Is the manuscript presented in an intelligible fashion and written in standard English?

Reviewer #1: Yes

Reviewer #2: Yes

6. Review Comments to the Author

Reviewer #1: (No Response)

Reviewer #2: All comments addressed - thanks

7. PLOS authors have the option to publish the peer review history of their article (what does this mean?). If published, this will include your full peer review and any attached files.

**Do you want your identity to be public for this peer review?** For information about this choice, including consent withdrawal, please see our Privacy Policy.

Reviewer #1: No

Reviewer #2: No
